# Magnetic Sensor Angle Adjustment to Improve Corrosion under Insulation Detection

**DOI:** 10.3390/s24030797

**Published:** 2024-01-25

**Authors:** Joseph Bailey, Gideon J. Gouws, Nicholas Long

**Affiliations:** 1Robinson Research Institute, Victoria University of Wellington, P.O. Box 33436, Lower Hutt 5046, New Zealand; joseph.bailey@vuw.ac.nz; 2School of Engineering and Computer Science, Victoria University of Wellington, P.O. Box 600, Wellington 6012, New Zealand; gideon.gouws@vuw.ac.nz

**Keywords:** eddy current testing, magnetic sensors, corrosion under insulation

## Abstract

A large portion of the pipe infrastructure used in the chemical processing industry is susceptible to corrosion under insulation (CUI). Eddy current-based magnetic sensing is one of the methods that can be used as an early detector of this corrosion. However, the large sensor-to-pipe distances used in this method, due to the presence of insulation, limits the sensitivity to corrosion. This paper will describe the development of instrumentation and methods based on eddy current sensing with thin-film magnetic sensors. In particular, it focuses on the influence of the sensor angle relative to the radial magnetic field. The influence of this parameter on the amplitude of the measured signal was investigated by both finite element simulations and experimental observations. The measured magnetic field was found to be highly sensitive to small changes in sensor angle, with the estimated depth of a defect changing at a rate of 11.2 mm/degree of sensor rotation for small angles. It is also shown that a sensor aligned with the radial direction should be avoided, with an optimal sensor angle between 0.5 and 4 degrees. With the sensor in this angle range, the simulations have shown it should be possible to resolve the depth of corrosion to a resolution of 0.1 mm.

## 1. Introduction

Insulated steel pipes are used widely throughout the chemical processing industry to transport steam and other hot gases or liquids, with many thousands of kilometres installed worldwide. If these pipes rupture due to undetected corrosion, this can create a health and safety incident. This would cause an unplanned system shutdown with significant economic impact [1].

A major source of this corrosion is “corrosion under insulation” (CUI). This occurs where water vapor or liquid water penetrates the protective pipe cover and condenses on the pipe surface. This can cause the steel surface to corrode, with no external visual indication until the pipe ruptures [2]. Several technologies have been investigated to enable an inspection system for the early detection of corrosion without the need for removing the insulation [3]. These solutions include ultrasound-based systems [4,5], microwave moisture detections [6], X-ray systems [7,8], magneto-strictive systems [9] and eddy current systems [10,11,12], with a review of giant magnetoresistance (GMR) magnetic sensor-based systems in [13]. In our previous work [14,15,16], a promising eddy current system was developed that uses a unique configuration where the coil encircles the pipe. This coil configuration was then combined with thin-film magnetic sensors such as GMR or tunnelling magnetoresistance sensors (TMR) to measure the axial, radial and tangential magnetic fields to the pipe.

The basic operation of an eddy current system is shown in Figure 1, with the magnetic field, generated by a coil, inducing an eddy current in the test object. When a defect such as corrosion is present in the test object, the eddy currents are distorted. This leads to distortion of the magnetic field produced by the eddy current. This distortion is measured either through the change in voltage generated in a sense coil or through another magnetic field sensor.

When testing for CUI the sensor and excitation coil need to be placed a significant distance from the surface of the pipe. This is due to the thickness of the thermal insulation, which can range from 25 mm to 100 mm. This distance, called the sensor lift-off, introduces two challenges. The first is due to the magnetic field strength decreasing with the cube of the distance from a solenoid coil. This results in a rapid decrease in the magnitude of the eddy current and thus the resulting magnetic field at the sensor with increasing lift-off. Secondly, the magnetic field tends to spread out with increasing distance from the source, resulting in a decrease in the spatial resolution of the system.

The system we have developed changes the standard excitation coil design to instead wrap the coil around the pipe [5,6,7] and blind trials have shown the promise of this system [18,19]. This configuration reduces the field decay with distance, resulting in higher eddy currents for a given power level. This enables the system to detect corrosion with a higher sensitivity at the large sensor to pipe distance present in the CUI application. In addition, instead of using a sense coil, thin-film magneto resistance sensors were used to measure the magnetic field at higher spatial resolution. This configuration does introduce some limitations in where it can be applied, as the coil needs to encompass the pipe. This means that areas with obstacles on the pipe such as supports cannot be tested. However, the coil can be demounted and assembled onto the pipe, which allows any obstacle free area of the pipe to be tested. This configuration would be suitable for the high-speed testing of straight pipes as the full circumference can be tested simultaneously at a high resolution and sensitivity. A slower tool would then need to be used in areas of the pipe with obstacles. Here, we present further analysis and experiments to improve the sensitivity of this system. The basic setup is shown in Figure 2.

While defect detection was achieved in our previous work [5,6,7], there was significant variation in the measured magnetic field not anticipated by simulations. This increased errors in detection and excluded the possibility of determining the defect depth from the results. It is hypothesized that a significant source of unexpected variations in the measured magnetic field is due to both the non-ideal placement as well as the non-ideal performance of a sensor. The work presented here will explore this hypothesis, extending the previous work by investigating the effects of the placement of the sensor on the measured magnetic field and its influence on defect detection.

The eddy current system used in this work relies on accurately measuring the radial magnetic field components to detect and determine the size of corrosion. In the ideal case of a completely uniform pipe with infinite length, constant permeability and conductivity values, the radial magnetic field is zero at any point along the line shown in Figure 3. This means a radial sensor placed on this line should measure zero field.

When there is an area of corrosion, the eddy current is distorted due to the rapid change in conductivity, changing the eddy current distribution. This distorts the eddy current so it no longer flows in a strictly circumferential direction as in the ideal case, and magnetic fields will be generated in the radial and tangential direction.

It follows from this idealized model that to improve defect detection, the induced eddy current should be increased to the maximum achievable, as this would produce the largest radial and tangential fields when corrosion is present. This would create a clear distinction between a corrosion-free pipe with no radial or tangential field compared to a pipe with corrosion, with a large radial and tangential field. This differential is important, as the sensor lift-off increases with thicker thermal insulation on the pipe, and the absolute field drops rapidly.

To maximize the induced eddy current, the amplitude of the exciting magnetic field needs to be maximized. A system could be designed that could generate the maximum field of 0.4 T that the sensor can tolerate [20]. However, this would require significant power and a cooling design. Due to practical limitations introduced by the sensor used in this work, an excitation current of 300 A was used, with a nominal axial magnetic field of ±1.5 mT.

One of the limitations of these TMR sensors is that they will possess non-zero cross-sensitivity. This means that a sensor that, for example, should measure a zero radial field will show a non-zero measurement when exposed to a strong axial field. Cross-sensitivity values are not typically reported for commercial TMR sensors, so it was determined experimentally and reported in the Experimental Design section of this paper.

In addition to this cross-sensitivity, a sensor cannot be practically placed so that it is exactly aligned radially or tangentially to the excitation coil and steel pipe. The typical placement accuracy of a sensor depends on the design, manufacture and assembly method of both the sensor and instrument system. Misalignment can be added throughout the process, starting with how the sensor die is packaged relative to the package pins, then the mounting of the package to the PCB, then finally, the mounting of the sensor PCB to the excitation coil. This paper investigates the effect of this misalignment on the ability to detect corrosion. For this work sensor misalignment between ±5 degrees are explored, with the actual misalignment angles determined in the Experimental Design section. In addition to uncontrolled misalignment, the potential for purposely misaligned sensors for improved corrosion detection is also investigated.

Past industry trials such as [19] have been interested in a corrosion range of 10% to 95% wall loss, which translates to corrosion depth between 1 and 9.5 mm for the 10-inch schedule 40 pipe used in this paper. This set the target minimum detection depth to 1 mm, while a desire to detect the rate of corrosion by measurement spaced at 1–2 years apart set the desired depth resolution to 0.1 mm. The value was reached by the data presented in [21,22], where corrosion rates of between 0 and 0.5 mm per year have been reported.

The sensor used for the experimental work presented in this paper is a tunnelling magnet resistance sensor, the TMR2001, from MultiDimension Technology Co., Ltd., Zhangjiagang, China. This sensor has a linear range of ±0.5 mT in air, a maximum tolerated field of 0.4 T and sensitivity of 8 mV/V/Oe [20].

The results in this paper are presented in the following two sections. In Section 2, FEM simulations using Opera™ [23] enabled the rapid assessment of a wide range of parameters. Section 3 describes the experimental system, which allows effects not captured in the simulations to be identified and assessed. The outcomes of these two approaches and the prospects for the system effectiveness are then discussed.

## 2. Finite Element Modelling Simulations

### 2.1. FEM Simulation Approach

A 2D axis symmetrical simulation of a 3000 mm long pipe with an outside radius of 160 mm and wall thickness of 10 mm was used. The Opera™ Harmonic EM solver, using non-linear material properties for the permeability, was used to solve the FEM simulations. The equations solved for this FEM simulation are detailed in the Opera Reference Manual [24], starting with Maxwell’s equations. The input driving source to the simulation was a current density of 1.2 A/mm^2^ in the excitation coil, resulting in a 300 A sinusoidal current at 10 Hz. A cylindrical boundary, with boundaries sufficiently distant to have a minimal effect, was used for the simulation. The boundary condition for the magnetic field was set to tangential for the radial boundary and perpendicular for the axial boundaries. The steel B-H curve was sourced for the Opera™ 2023 database [23], with the average value for mild steel used. The conductivity of the steel was set to 1.43 × 10^7^ S/m.

Due to the rotational symmetry of the model, defects introduced to the simulation are finite in the radial and *z* direction, but exist for all φ ϵ [0, 2π]. In addition, the rotational symmetry means only the axial and radial field are modelled and the tangential field is not captured by this simulation. As such, only the radial and axial field will be considered in the work presented here.

The base FEM model used in this work is shown in Figure 3. In this model, the effect of sensor misalignment on the measured radial and axial magnetic fields can be assessed. A high-density mesh was used in the excitation coil, the section of pipe below the excitation coil and in an area around the measurement location. The field was evaluated at two measurement points P and Q. Point Q is between the coil and pipe, the location that is expected to have the best response. Point P is outside the coil, the location of the sensor in the experimental work, to allow the easy adjustment of the sensor angle.

The mesh was optimized by reducing the mesh size in the high-density location until an acceptable balance between simulation time and accuracy was achieved. This balance was achieved with a mesh size of 0.5 mm, resulting in a solve time of less than 60 s and mesh error caused by discretization of up to 20 nT on the radial field and 0.1% on the axial. The resulting mesh that was used throughout this study has 400,000 triangular elements, with a mixture of structured and unstructured meshing.

### 2.2. FEM Sensor Angle Effect

The effect of sensor rotation from alignment with the radial direction was investigated with and without simulated corrosion present. From these results, the effects of both intentional and unintentional misalignment of the sensors are assessed, along with the effect of dynamic misalignment, where *θ*, the mis-alignment angle, changes with time.

The base simulation, described in Section 2.1, was used to simulate the effect of rotating the sensor ±2 degrees from radial alignment with the results shown in Figure 4. This shows that the radial field is highly sensitive to sensor angle, with 20 µT/deg sensitivity at small angles. This value is three orders of magnitude larger than the 20 nT accuracy achieved by the simulation of the radial field. The dependence of the axial field sensitivity was also explored, but only showed a change of 0.05% over 2 degrees. This value is below the expected error level for the axial field of 0.1% determined in Section 2.1.

Having shown the significant effect of the sensor angle on the measured magnetic field, corrosion was added to the simulation to explore the effect of the sensor angle on detecting and sizing corrosion. The corrosion was simulated by thinning the pipe wall in a section. For the initial simulation, a 100 mm long section was reduced by 5 mm. The simulation is made up of a set of static simulations where the location of the coil and measurement point P and Q are moved axially in 5 mm steps to approximate the measurement process of scanning a pipe. This results in a plot of the radial field at points P and Q along the length of the pipe. There are two sources for how sensor angle error can occur. A static error will result from any misalignment between the sensor and package during the fabrication process or during device assembly. It is estimated that there is a potential for up to 5 degrees of misalignment from this source. The second source of error is due to movement of the sensor during measurement. This error would be of a much lower magnitude and would depend on the environment and sensor mounting design.

The static misalignment of the sensor is simulated first, with Figure 5 showing how the measured radial field changes with sensor angle as it is moved over the defect. The changes to the radial field are anticipated with large increases in the background field. However, the shape of the distortion in the magnetic field magnitude has an interesting response to the sensor angle, with a rapid change occurring in the first 0.5 degrees of rotation. The distortion transitions from two peaks at zero degrees to a peak and a valley at 0.4 degrees and from then on there is little change in the distortion of the magnetic field.

There are two metrics of importance when measuring the distortions in magnetic field generated by corrosion. The first is the peak-to-peak value of the distortion. This is determined by finding the difference between the maximum and minimum magnetic field magnitude within an envelope of *z* positions that starts 200 mm before the defect and ends 200 mm after the defect. This is important as it sets the signal level that can be used to identify corrosion. Second is the relative amplitude, calculated by taking the peak-to-peak value and dividing it by the background magnetic field magnitude. This background value is found as the average value in an envelope either side of the defect. This envelope starts 200 mm from the defect and ends 300 mm from the defect. The relative amplitude is important as it determines what change in peak-to-peak value can be resolved for a measurement system with a given dynamic range. This allows for comparison of the measurability of signal independent of the measurement system used.

The effect of sensor angle on the peak-to-peak value and relative amplitude is shown in Figure 6 and Figure 7. Both metrics show a high dependence on sensor angle; however, there is a competing optimal sensor angle, with a minimum peak-to-peak value occurring at the 0 degrees while the maximum relative amplitude occurs at 0 degrees. These competing values suggest an optimal sensor angle may be found between 0.3 and 1 degrees depending on the dynamic range of the analog-to-digital converter used.

In addition to the static sensor angle, the effect of the dynamic sensor angle was also simulated. This was achieved by setting the sensor angle to a random number with a gaussian distribution with a mean of 0 at each *z* step of the simulation. Figure 8 shows the results for σ (standard deviation) from 0 through to 0.1 degrees. This shows that the large axial field means that the sensor mounting needs to be rigid, as anything above 0.01 degrees of movement can introduce noise that would need significant data processing to determine the peak-to-peak value for the magnetic field distortion.

### 2.3. Defect Size Effects

To contextualize the magnitude of the effects caused by sensor rotation, a set of simulations were completed to assess the effect of defect size on the peak-to-peak signal generated in the radial magnetic field. The simulation setup shown, in Figure 3, was used, with defect depths from 1 to 5 mm and lengths from 50 to 150 mm and the sensor angle set to zero degrees. An example of the radial magnetic field change for a 100 mm long defect is shown in Figure 9. The change in peak-to-peak value for all the defect sizes was then simulated, and is shown in Figure 10.

To be a useful defect detection system, the system must identify both the presence of corrosion and its severity. As discussed in the introduction a minimum defect depth of 1 mm with a resolution of 0.1 mm is the target for this system. These simulations have shown a signal with peak radial magnetic magnitude of 1.2 µT is expected for 1 mm depth with a gradient between 1.1 to 3.1 µT/mm depending on the location of the sensor and the length of the defect. This means any changes in the magnetic field from unknown sources larger than 0.11 µT are relevant and will increase the uncertainty in the measurements.

### 2.4. FEM Summary

The simulations have shown that, if a sensor angle of 0 degrees is used, any small changes in sensor angle will affect the ability to determine defect depth from the peak-to-peak radial magnetic field. The peak-to-peak value changed at a rate of 23.5 µT/deg inside or 22.8 µT/deg outside the coil. Combining this angle sensitivity to the depth sensitivity shown in Figure 10 a value for the depth error as function of sensor angle can be determine. This is 11.2 mm/deg at the inside sensor location and 20.7 mm/deg at the outside sensor location.

This high sensitivity of sensor angle near zero along with the results shown in Figure 6, and Figure 7 shows that an ideal sensor angle of around 1 degree is needed to reduce sensor angle sensitivity while still producing a high relative amplitude for defect distortion. In addition to informing the determination of a preferred sensor angle, the simulations have also highlighted the importance of rigidity of the sensor mounting with any rotation on the sensor above 0.01 degrees likely to introduce detrimental noise into the measurement. This noise would affect the measurement of peak-to-peak signal from the defect and signal processing would be needed to find this value to within 0.11 µT accuracy that the simulation show is needed to determine defect depth to a resolution of 0.1 mm.

The simulations showed the potential for the measurement of the magnetic field that would enable the depth of corrosion to be determined to a resolution of 0.1 mm. However, the simulations did not model the effect of parameters such as the cross-sensitivity of the sensor and non-uniform pipe properties. Section 3 will look to address these additional sources of error using experiments.

## 3. Experimental Design and Results

### 3.1. Experiment Setup

The integration of the sensor system with the pipe under test and the motion system allowed the construction of an experimental NDT system shown in Figure 11. This should also allow for the measurement of parameters, such as non-axis symmetrical defects, pipe non-uniformity and sensor cross-sensitivity, that could not be included in the FEM simulations.

#### 3.1.1. Sensor

The sensor used for this experiment section is the TMR2001 from MultiDimension [22]. TMR sensors are known for their low cross-sensitivity [25] although the actual cross-sensitivity of this device is not reported by the manufacturer. This is a critical value for the work presented here, as the large magnetic field perpendicular to the measured magnetic field will be picked up by any cross-sensitivity present in the sensor. The other critical unknown is the angle of the sensor compared to the reference edge of the PCB it is mounted on, which should nominally be zero degrees. However, due to the assembly process, this is expected to vary by up to ±2 degrees. This is expected to introduce a large background field as shown by the FEM simulations in the previous section.

To address these sensor error issues, a calibration process is needed to determine the cross-sensitivity and sensor angle. There are many calibration processes that have been developed for magnetic sensors. These include a calibration process for the measurement of the magnetic rotation position in [26], and the calibration of a magnetometer for space applications are discussed in [27,28]. A 3D Helmholtz coil system is used in [28], where a known magnetic field at a range of angles and strengths is used for calibration. Ref. [27] presents a system that only requires a single coil. A calibration process is presented in [29] that addresses the full calibration of magnetic sensors, which includes the scaler, sensor frame, mounting frame, and anatomical frame calibration.

To calibrate the cross-sensitivity and sensor angle for this application, a calibration system that uses a solenoid coil that can produce magnetic fields ranging ±17.4 mT was used. The sensor can be mounted in the centre of the coil at a range of fixed angles between ±5 degrees relative to the axis of the coil.

The slope of the sensor sensitivity was calculated as the ratio between the measured field and applied field. This was calculated for each sensor angle in Figure 12. A linear fit was applied to the positive and negative slope of each sensor curve. The angle of misalignment is the point where the two fit lines intersect. The field measured at this angle of misalignment is the cross-sensitivity of the sensor, with the cross-sensitivity defined as the percentage of the field perpendicular to the sensing axis that is added to the measured field. These values are summarized in Table 1 for a set of four identical sensor units to show the typical variation due to unit assembly variation. These values were compared against manufacturer’s values when available and were all within the expected range. 

The values measured were all within expected values and the measured value of the cross-sensitivity was relatively low. However, due to the perpendicular field being potentially 3 orders of magnitude larger than the field of interest, any changes in the cross-sensitivity or the perpendicular field were likely to have a measurable effect.

In addition to error introduced by the cross-sensitivity and sensor misalignment, the measurement system has a background noise level with a standard deviation of 0.022 µT. This is well below the target 0.11 µT needed to detect 0.1 mm changes in defect depth.

#### 3.1.2. Pipe Samples

A wide range of pipes are used in industry, with variations in size and material composition depending on the situation. We have chosen to use a nominal 10” schedule 40 seamless Linepipe Grade B ASTM A106 [29]. This was the closest we could find to a typical pipe, used in industry, that could be purchased in small quantities.

The standard allows for variation of +2.4 mm, −0.8 mm in the outside diameter and ±1.35 mm on the inside. The nominal wall thickness was 9.27 mm with a lower bound of 8.11 mm. A 3 m pipe was used with the testing focused on the middle 1 m of pipe to reduce any effects for the ends of the pipe. The wall thickness was measured with an ultrasound probe and variation between 9 and 9.8 mm was observed around the circumference of the pipe.

The pipe was first tested as delivered, after which a set of defects of various sizes and two geometries were machined into the pipe to simulate corrosion. These defects were of the geometry shown in Figure 13. The box defects produce clearly defined edges and slot defects which have a clearly defined *z* edge but a more gradual change in the circumferential direction. The important property to note with these defects is that as they have a flat bottom, the wall loss varies over the defect. In total, 10 defects were machined into the pipe. A summary of the defects size and type is in Table 2. 

#### 3.1.3. Motion and Excitation System

The final part of the experimental system is a translation and rotation system to scan the magnetic sensor over the surface of the pipe and an excitation system to induce eddy currents in the pipe. The system consists of a pair of linear stages on which the excitation coil and magnetic sensor are mounted and moved down the length of the pipe. A large servo motor then rotates the pipe so that combined with the linear stages the magnetic sensor can be scanned over the full surface of the pipe.

The eddy current is induced at a known frequency and phase by using a signal generator which drives a high-power amplifier. This in turn drives current into the primary of a toroidal transformer. Finally, the secondary of the transformer is a single copper loop producing a current of 300 A, creating the magnetic field to induce eddy currents in the pipe. To maintain a constant magnetic field, the current in the copper loop is monitored with an open-loop Hall effect sensor. This is used to modify the amplitude of the voltage from the signal generator to maintain constant excitation current to within ± 0.5 A at 300 A. This results in a field stability of ±0.17%

To characterize the ability of the translation and rotation system to position the sensor, a linear potentiometer was used to measure the lift-off distance from the coil to the pipe surface, *L*. The movement system was used to map a 600 mm × 360 deg section of the pipe. Figure 14 shows the result of this measurement with a maximum variation in *L* of 2 mm over the surface. There appeared to be two types of variation occurring, a constant increase in *z* and a more random variation on top of this. This suggests two sources causing the variation. The constant increase in *z* of 1 mm/m is likely due to a misalignment between the *z* stages and pipe axis. The rest of the variation can be accounted for by variation in the diameter of the pipe with 2 mm within the variation allowed by manufacturing standards. This variation in distance is a likely source of magnetic field distortion not included in the simulations.

### 3.2. Experimental Results

#### 3.2.1. Effect of Sensor Angle and Position Relative to Excitation Coil

The simulations from Section 2.1 showed a radial magnetic field sensor to be highly sensitive to the placement of the sensor and this relationship was also experimentally determined. For the experimental work, sensor unit 4 from Table 1 was mounted on a rotation stage. This setup allowed the measurement of the sensitivity to rotation in the *r*-*z* plane.

The sensitivity to rotation was assessed by positioning the sensor in the axial centre of the excitation coil with an error of ±0.2 mm via measurements with callipers. This was set as the nominal *z* = 0 point. From the sensor calibration in Section 3.1.1 the sensor misalignment of 1.7 degrees was expected so the sensor was rotated from 0 to 1.8 degrees to capture the expected minimum. This measured magnetic field with sensor angle is shown in Figure 15.

The minimum measured field was found between 1.26 degrees and 1.32 degrees. This shows the sensor was perpendicular at 1.29 degrees. The 0.4 degree change from the calibration represents the mounting error introduced when attached to the rotational stage. The small flat region with a field of 2.5 µT is the region dominated by the cross-sensitivity signal from the axial field. The main axial field was measured at 1200 µT, which gave a cross-sensitivity of 0.2% for this sensor, close to what was expected from the calibration.

Fitting a linear slope to each side of the V curve gives an angular sensitivity of 18.8 µT/° or to generalize 1.6%/° of background for small angles. This aligns well with results expected from the simulation of 20 µT/° and 1.3%/°.

#### 3.2.2. Defect-Free Pipe Mapping

Using sensor unit 4 the radial magnetic field over a 600 mm × 360 degree section of the pipe was mapped before defects were introduced with the sensor positioned at the minimum angle determined in Section 3.2.1. The map of this magnetic field is shown in Figure 16. This shows there was a significant magnetic field distortion up to 40 µT generated by the pipe before defects were introduced. There also appeared to be some structure to the observed magnetic field variations, for example, a ridge of higher magnetic field spiralling around the pipe with a pitch of 600 mm. This does not match the variation in lift-off mapped in Figure 14. It is suggested that changes in the conductivity or permeability of the steel could have been the cause of these variations. However, to confirm these, further experiments would be needed.

The as-delivered pipe had large variations in the radial magnetic field, as shown in Figure 16. As the field changed with the position in the *z*-*ф* plane, unlike in the simulations, we expected changing the sensor angle would cause a varying change to the measurement of *B*. This was different from the simulations previously presented, where *B* was constant for all positions for a pipe without defects. The interaction of varying *B* and sensor angles would increase the uncertainty in defect identification and sizing.

To investigate the issue, the mapping of the pipe was repeated at a range of sensor angles for a section of pipe 310 mm long. From each magnetic map, the average field across the whole scan and maximum change in field across the whole scan was recorded. In addition, the prominent peak in the magnetic field at 200 mm and 150 degrees (peak 1) was used to assess how the sensor angle affects a specific variation in the magnetic field. The peak-to-peak metric used for defects could not be used here as just a single peak was being assessed. The metric used in this case was the relative peak value, which is defined as the difference between the average magnetic field across whole map and maximum value of the peak. The relative peak value was determined for each sensor angle. These values are summarized in Table 3.

The results showed that changes in sensor angles below ±1 degree do not produce measurable effects with this system. However, a change of two degrees or more did produce a measurable change across all parameters apart from the relative peak value. The average and peak 1 maximum field increased with increasing angle, as expected from the simulations. However, the relative peak value for peak 1 did not increase uniformly with the sensor angle. Instead, it stayed relatively stable for all sensor angles. These intrinsic variations in radial magnetic field, caused by non-uniformity in the pipe, would not stop defect detection but would likely require additional signal processing to remove their effects.

#### 3.2.3. Map of Pipe with Defects

Once testing of the pipe, as delivered, was completed, a set of 10 defects were machined into the pipe, as described in Section 3.1.2. The radial magnetic field of the pipe was then mapped, as shown in Figure 17. In general, each of the defects produced a pair of peaks on the axial edges of the defect, with the size of these peaks increasing with an increased defect volume as predicted from the simulations. However, the background variations in some areas were larger than those produced by the defects and the smaller defects were obscured by the local changes in magnetic field caused by the pipe.

To understand how the sensor angle affects the defect signal, defect number 5 (ref Table 2) was mapped, with sensor angles ranging from −2 to 4 degrees. An example of these maps with a sensor angle of zero degrees is shown in Figure 18.

To assess the effect of sensor angle on the magnetic field above defect 5, the metrics used to assess the simulations in Section 2.2 were used. Figure 19 shows the effect on the peak-to-peak amplitude and Figure 20 the effect on the relative peak-to-peak amplitude. This result matches what was observed in simulation, with a steep valley or peak at 0 degrees for each metric, as shown in Figure 6 and Figure 7.

The cause of this rapid change around the zero-degree sensor angle was illustrated by plotting the magnetic field for the sensor at −2 and 4 degrees (Figure 21). What is important to note is rotating the sensor changed a peak in the magnetic field to a valley and vice versa. This means there was a sensor angle between −2 and 4 degrees where each peak value became equal to the background level on its way to transitioning to a valley. If this transition for both peaks occurred at the same angle, then the signal would drop to zero. However, for a small set of angles, the result was two peaks in the magnetic field and no valleys, as seen in Figure 18. It is this phenomenon that causes the rapid change in the peak-to-peak and relative peak-to-peak value around zero degrees.

## 4. Discussion

The simulations showed that the measured magnetic field was highly sensitive to the angle of the sensor, with a sensitivity of 20 µT/°. This high sensitivity to the sensor angle means any movement in the sensor during the measurement will introduce noise to the measurement. To achieve a low-noise measurement, the system needs to have the stiffness to hold the sensor steady, with less than 0.01 degrees of movement.

The simulations and experiments showed that the peak-to-peak signal drops rapidly as the sensor angle approaches zero degrees, while the relative amplitude increases rapidly as a sensor angle of zero is approached. This result shows that sensor angles of zero degrees should be avoided in the system design due to the instability at this point.

To determine which sensor angle is optimal, the design of the measurement system must be considered, in particular the dynamic range. A larger dynamic range allows larger sensor angles, as this maintains the ability to resolve small changes on a large background that occurs when the sensor rotates. If a large enough dynamic range cannot be achieved for sensors with an angle between 2 and 4 degrees, then sensor angle between 0.5 and 1.5 degrees should be chosen, balancing the need for a large signal-to-background level and large peak-to-peak values.

The simulations have also shown that amplitude of the distortion in the magnetic field is correlated with the depth of the defect. In the case of the simulation, this is 1.1 to 3.1 µT/mm. The experimental variation in the magnetic field measurement, when a lock-in measurement is used, has a standard deviation of 0.022 µT. This means it would be feasible to measure changes of 0.1 mm of defect if all other factors were controlled. However, more data are required on the effect of the defect shape on the peak amplitude to enable the defect depth prediction.

The experimental work showed that sensor angle alignment when soldered by hand can reach at least 2 degrees, proving the effects investigated in the ±2-degree range in the simulations are relevant. The testing on pipe as delivered from the factory showed it created large distortion in the radial magnetic field. Despite this, all but the smallest 2 defects could be clearly identified in the magnetic field maps, with the general trend of large defects producing the larger peaks observed.

Changing the sensor angle in the experiments showed the same effect predicted by the simulations. As seen in Figure 19 and Figure 20, there is a rapid change in peak-to-peak and relative peak-to-peak values at a zero-degree sensor angle. This confirms that if sensor angles between ±0.5 degrees are avoided, defects can clearly be identified, as in Figure 21.

## 5. Conclusions

We have shown an eddy current system that can detect corrosion under insulation in steel pipes. The system uses a coil that encircles the pipe and TMR sensors to measure distortion in the magnetic field in the radial direction. The important system parameter of the sensor alignment with the radial direction was investigated and was shown to greatly affect both the background magnetic field as well as the amplitude of distortions generated by defects. Experiments have shown that a sensor angle of zero degrees should be avoided and sensor angles of between 0.5 and 4 degrees should be used depending on the instrumentation design.

Large variations in the background field were observed in the experimental results. Simulations showed that the relationship between the defect depth and magnetic field peak-to-peak magnitude change with defect length. If these two factors can be understood and controlled, the system has shown the potential to determine the depth of defects approaching the targeted 0.1 mm for corrosion under insulation application.

## Figures and Tables

**Figure 1 sensors-24-00797-f001:**
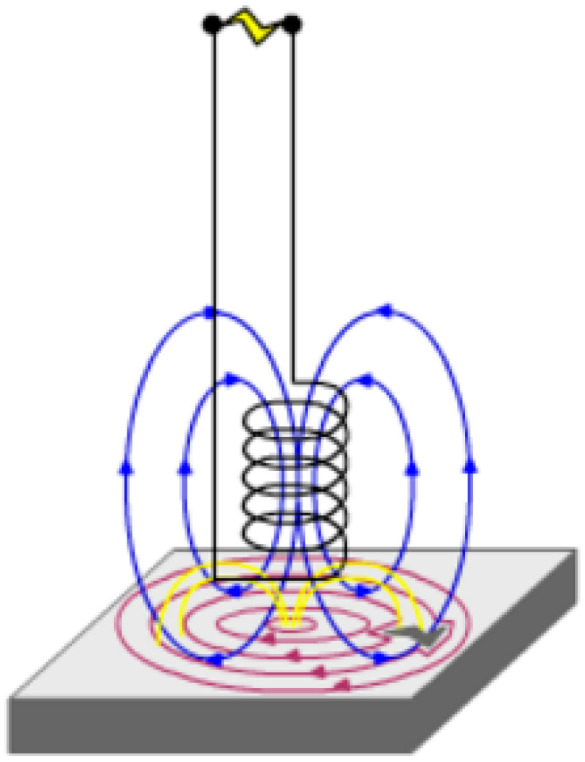
The principles of an eddy current testing system, where the blue lines show the magnetic field generated by the coil, the red lines show the eddy currents in the test object and the yellow lines indicate the magnetic field generated by the eddy currents [17].

**Figure 2 sensors-24-00797-f002:**
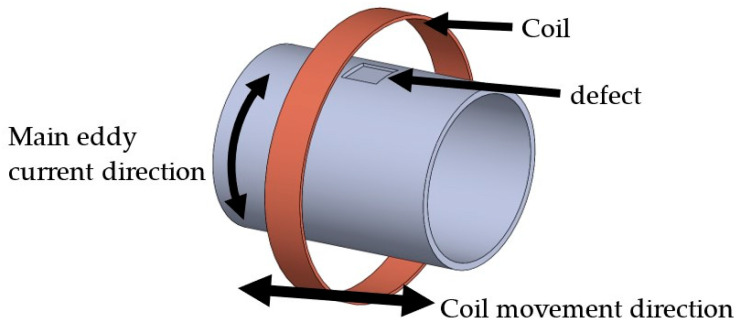
The basic configuration of the eddy current system is with the coil concentric to the pipe and the main magnetic field aligned with the pipe’s axial direction.

**Figure 3 sensors-24-00797-f003:**
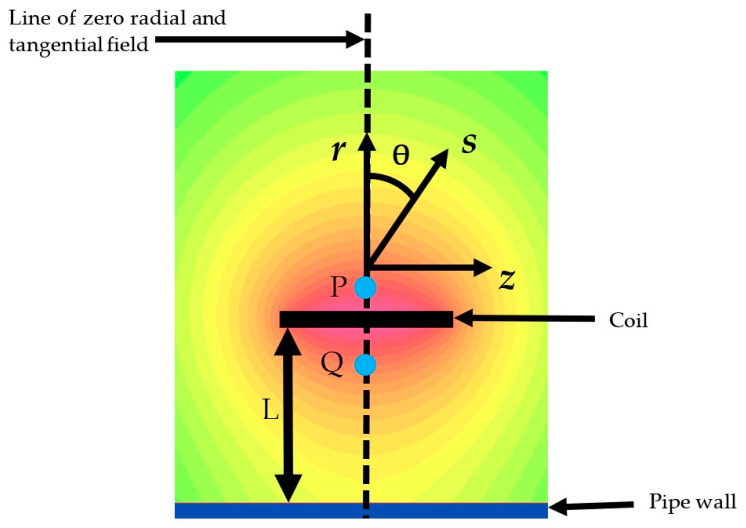
Axisymmetric layout of pipe simulation showing magnetic field with a line of zero radial field for the ideal case of uniform pipe and coil. P, Q are the points where magnetic field is evaluated. The radial direction is ***r***, the axial direction ***z***, sensitivity axis of sensor aligned to ***s***, and θ is angle between ***r*** and ***s***. L is the lift-off distance between coil and pipe surface.

**Figure 4 sensors-24-00797-f004:**
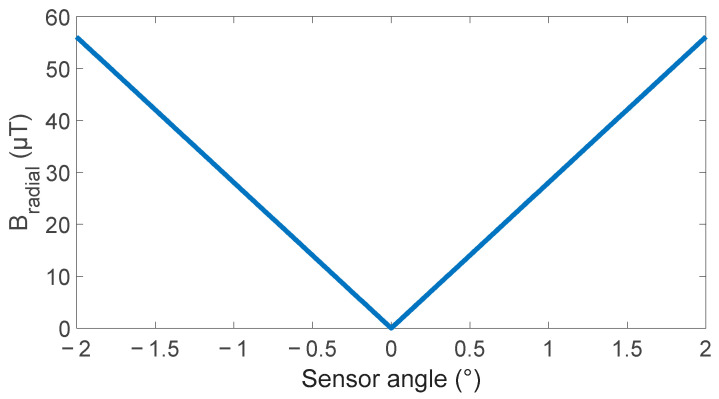
Simulated effect of sensor angle on magnetic field measured by a radial magnetic sensor.

**Figure 5 sensors-24-00797-f005:**
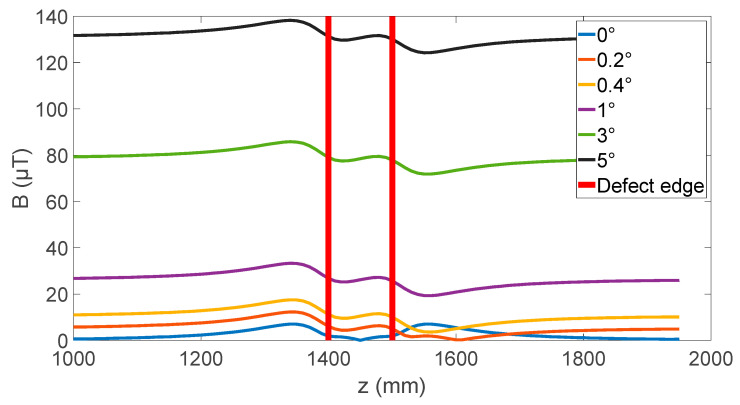
Effect of sensor angle on radial magnetic field at point P when moved over a 100 mm corrosion defect.

**Figure 6 sensors-24-00797-f006:**
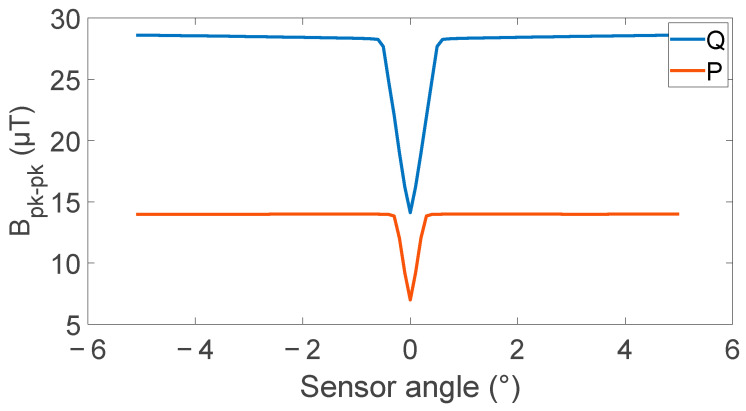
Effect of sensor angle on peak-to-peak defect signal generated at points P and Q by a 5 mm deep defect showing high sensitivity to sensor angles below 0.6 degrees, with a rate of change of 23.5 µT/° for the first 0.6 degrees on the inside and 22.8 µT/° for the first 0.3 degrees on the outside.

**Figure 7 sensors-24-00797-f007:**
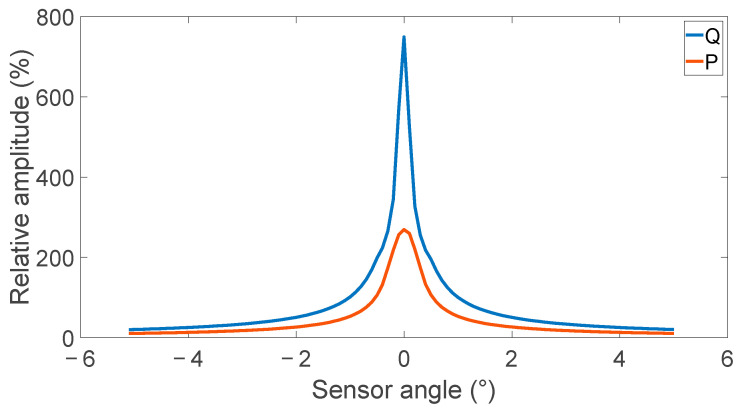
Effect of sensor angle on defect signal’s relative amplitude from the background field level showing a high-angle dependence for small angles at the measurement locations P and Q outside and inside the coil.

**Figure 8 sensors-24-00797-f008:**
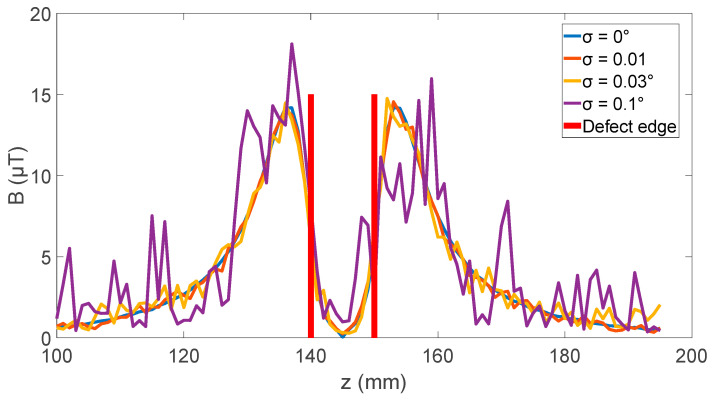
Effect of gaussian distributed sensor angle movement during measurement with a mean of zero degrees on the radial magnetic field measurement with an axial field of 1500 µT and peak radial field of 14 µT.

**Figure 9 sensors-24-00797-f009:**
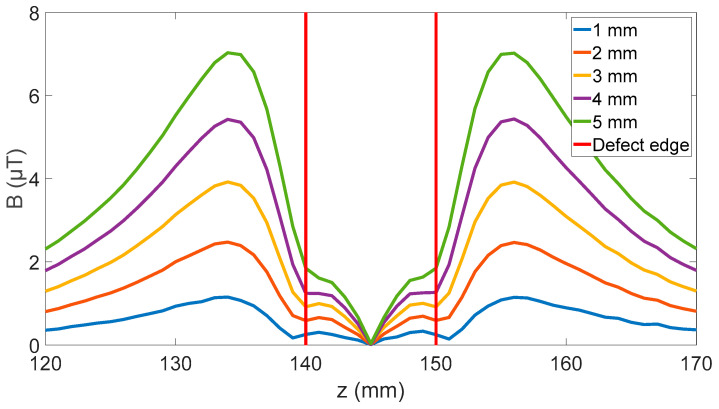
Effect of 150 mm defect with varying depths on radial magnetic field at P showing increasing effect until the peaks split. A wall thickness of 10 mm was used in the simulation.

**Figure 10 sensors-24-00797-f010:**
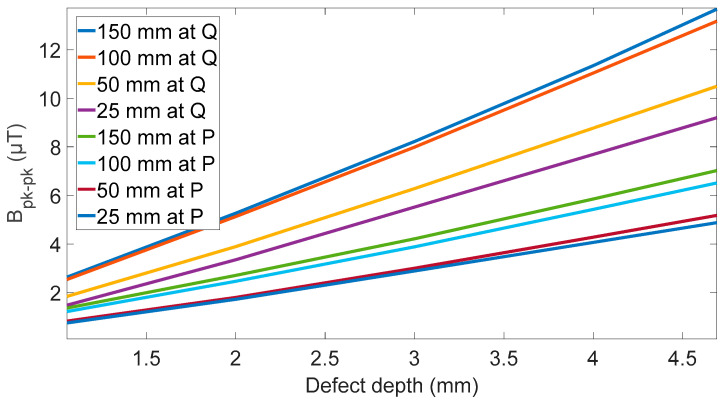
Effect of defect depth on different lengths of defects, showing a linear slope ranging from 1.1 to 1.5 µT/mm for outside sensor and 2.1 to 3.1 µT/mm for inside sensor.

**Figure 11 sensors-24-00797-f011:**
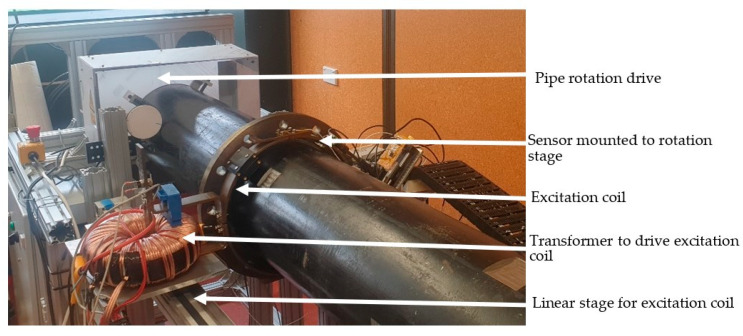
Photo of experimental setup with coil encircling pipe.

**Figure 12 sensors-24-00797-f012:**
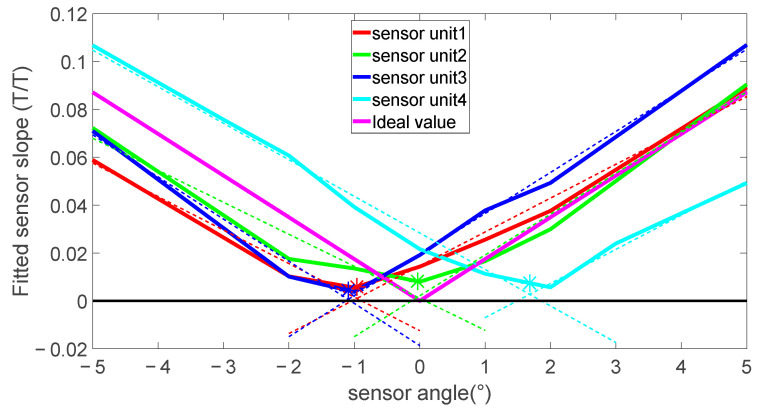
Sensor angle and cross-sensitivity calibration curves showing effect of the PCB angle on the sensor sensitivity slope.

**Figure 13 sensors-24-00797-f013:**
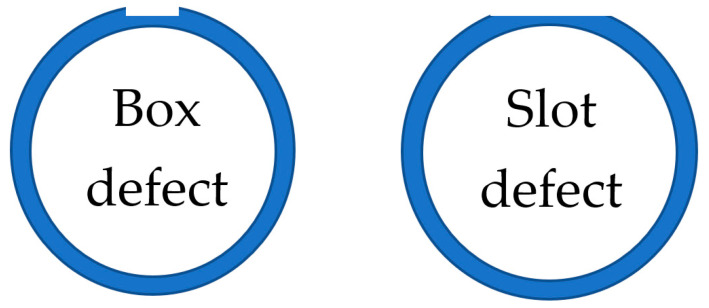
Cross-sectional view of pockets machined into pipe to represent corrosion.

**Figure 14 sensors-24-00797-f014:**
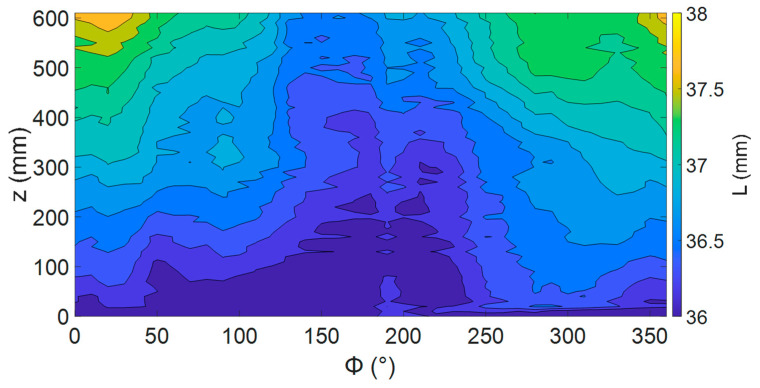
Change in sensor lift-off over the surface of the pipe in the test area.

**Figure 15 sensors-24-00797-f015:**
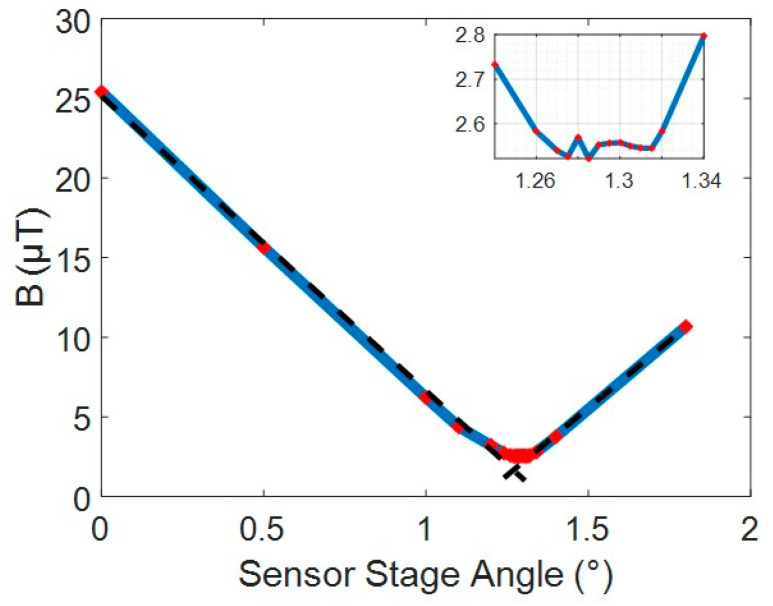
Effect of sensor rotation. Inset shows a detailed view of the turning point. Red diamonds show data points with the blue line showing the linear interpolation between points.

**Figure 16 sensors-24-00797-f016:**
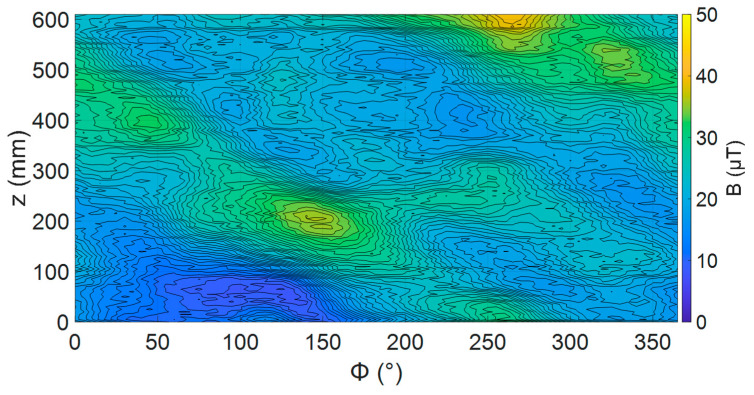
Radial magnetic field map over centre section of pipe as manufactured with sensor at zero degrees.

**Figure 17 sensors-24-00797-f017:**
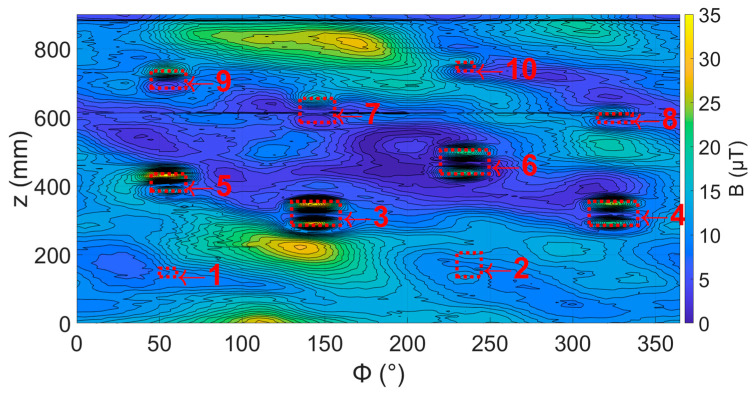
Map of radial magnetic field with sensor at 0 degrees over pipe with 10 defects introduced. Red boxes show the location of each defects 1 to 10 from Table 3.

**Figure 18 sensors-24-00797-f018:**
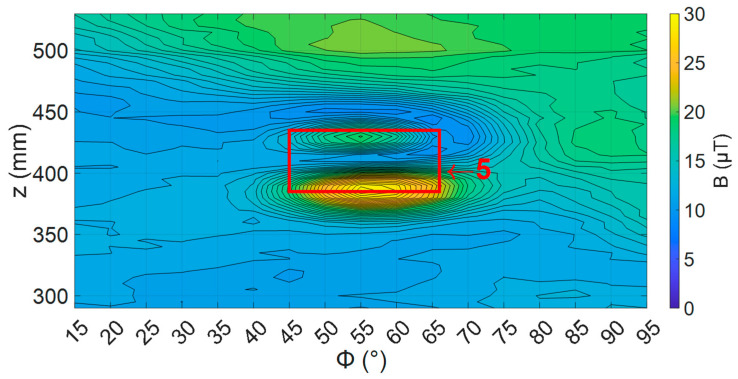
Defect 5 with sensor at zero degrees. Red box shows the locations of defect 5 from Table 3.

**Figure 19 sensors-24-00797-f019:**
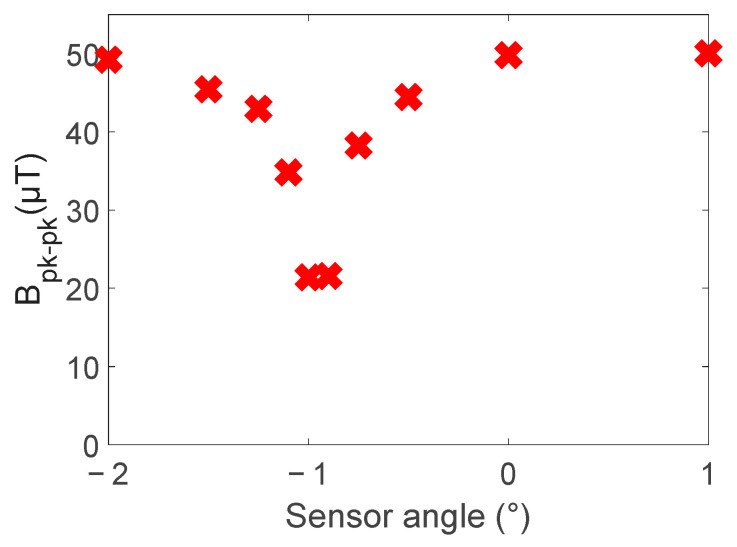
Effect of sensor angle on radial magnetic field magnitude peak-to-peak change with sensor angle for defect 5.

**Figure 20 sensors-24-00797-f020:**
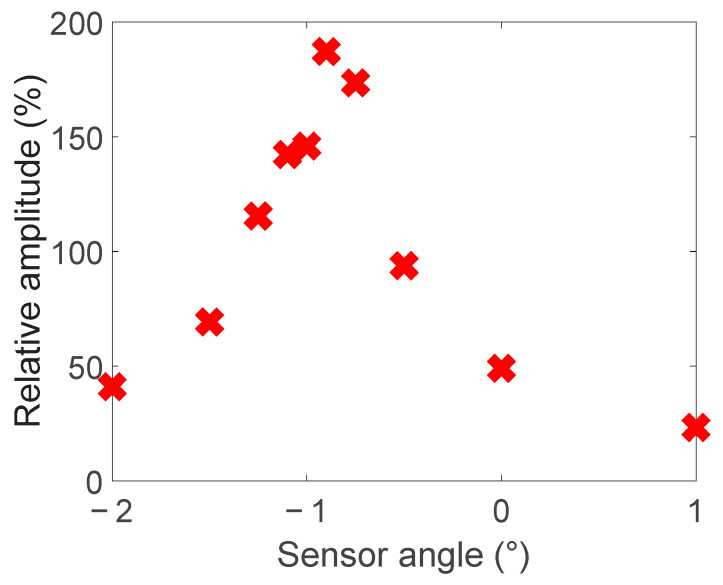
Effect of sensor angle on the relative amplitude of the peak-to-peak magnetic field variation.

**Figure 21 sensors-24-00797-f021:**
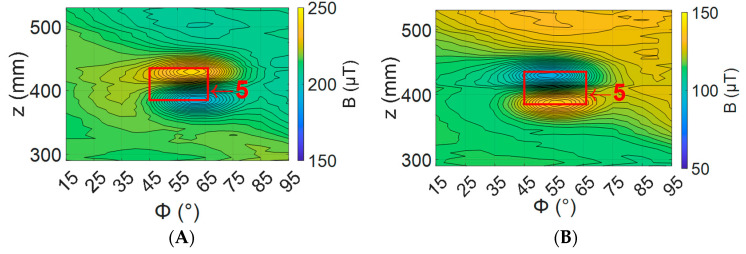
Effect of sensor rotation on magnetic field measured above a defect shown peak flipping. Red box shows the locations of defect 5 from Table 3. (**A**) Sensor at 4 degrees (**B**) Sensor at −2 degrees.

**Table 1 sensors-24-00797-t001:** Sensor attributes as determined in custom calibration system.

	Unit 1	Unit 2	Unit 3	Unit 4	Expected
Cross-sensitivity (%)	0.59	0.82	0.43	0.74	Unknown
Average field at zero current (µT)	−123.9	−104.8	−54.0	17.0	±375
Average sensitivity at 5° (mV/µT)	338.0	371.2	406.4	361	398.0
Sensitivity range at 5° (mV/µT)	3.3	9.0	2.1	25.7	Unknown
Estimated misalignment (°)	−0.9	0.0	−1.1	1.7	±2

**Table 2 sensors-24-00797-t002:** Defect machine into test pipe.

Defect ID Number	Axial Length (mm)	Radial Length (mm) or (°)	Max Depth (mm)	Defect Type
1	25	25, (10)	1.62	Box
2	70	35, (15.8)	1.21	Slot
3	70	70, (30)	7.42	Box
4	70	70, (30)	5.93	Box
5	50	50, (21)	7.48	Box
6	70	75, (31)	5	Slot
7	70	50, (21)	2.5	Slot
8	25	50, (21)	6.48	Box
9	50	50, (21)	3.48	Box
10	30	30, (13)	6.62	Box

**Table 3 sensors-24-00797-t003:** Summary of effect of sensor angle on measured magnetic field over pipe as delivered from factory.

Sensor Angle (°)	Average Field (μT)	Field Range (μT)	Peak 1 Maximum Field (μT)	Peak 1 Relative Peak Value (μT)
0	21	28.2	36.5	15
0.1	18.6	31.4	31.4	12.8
−1	14.9	34.3	23.0	8.1
1	14.5	26.9	23.0	8.5
2	133	55.4	148	15
5	280	51.9	293	13

## Data Availability

Data are contained within article.

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
