# Peer review of "Magnetic Sensor Angle Adjustment to Improve Corrosion under Insulation Detection"

_sensors, 2024, doi:10.3390/s24030797_

Round 1

Reviewer 1 Report

Comments and Suggestions for Authors

The influence of the sensor angle relative to the magnetic field was analyzed in the work. Such an eddy current sensor is used for external examination of pipelines to detect flaws and defects such as corrosion located under the insulation. The work is interesting and describes the examined issue in detail. However, I have a few comments.

1) Reviewing the work is difficult because in many places I see the message "Error! Reference source not found" (lines: 82, 154, 234, 247, 250, 311, 316, 349, 372, 386, 415, 436, 460, 461 , 466, 467, 478, 484, 522, 525).

2) If there is no photo of the FEM model mesh, please add information about it, e.g. what elements were used, how many elements the mesh had, etc.

3) The work should clearly explain the advantage of the proposed solution over the classic, single contact sensor with a core that can be placed on the pipe anywhere. Figure 2 shows that the coil must surround the pipe, so access to one end of the pipe is required (to install the coil), which is a certain limitation in practical pipeline inspections. In addition, any obstacle in the pipeline, such as a connector or handle, prevents the coil from moving further along the pipeline. It is worth discussing these topics at work, especially the limitations in using the presented method.

Minor comments.

4) Line 148: Shouldn't it be 1.43*10^7 S/M instead of 1.43*10^-7 S/M?

5) Line 59: "Eddy current" should probably be written in lower case, it is "eddy current".

6) Check your work for double spaces (too long spaces between words). Lines: 57, 58, 69, 102, 139.

Author Response

We thank the reviewers for their careful consideration of our manuscript and the useful suggestions made for improvements. Detailed responses to comments follow.

Reviewer 1

  • Reviewing the work is difficult because in many places I see the message "Error! Reference source not found" (lines: 82, 154, 234, 247, 250, 311, 316, 349, 372, 386, 415, 436, 460, 461 , 466, 467, 478, 484, 522, 525).

Yes, sorry, there appears to have been an issue with the cross-referencing in the Word document. We have replaced these links with text, so this should no longer be an issue. 

  • If there is no photo of the FEM model mesh, please add information about it, e.g. what elements were used, how many elements the mesh had, etc.

These details have been added in line 175 with a model with 400,000 triangular elements of a mix of structured and unstructured mesh. The rest of the model setup is included in the paragraph starting line 148.

  • The work should clearly explain the advantage of the proposed solution over the classic, single contact sensor with a core that can be placed on the pipe anywhere. Figure 2 shows that the coil must surround the pipe, so access to one end of the pipe is required (to install the coil), which is a certain limitation in practical pipeline inspections. In addition, any obstacle in the pipeline, such as a connector or handle, prevents the coil from moving further along the pipeline. It is worth discussing these topics at work, especially the limitations in using the presented method.

We have added some comments on this issue in the Introduction section, from the paragraph starting on line 62. Line access to the end of the pipe is not required, as the coil can be demounted and placed around the pipe. But the reviewer is correct, any connector such as a T junction will limit the detection method to operating at some distance along the pipe from the obstruction. This may be minimised using magnetic flux guiding but this topic is beyond the current paper. The system has two main benefits (1) the ability to measure the full circumference of the pipe in one measurement cycle, greatly speeding up the inspection; (2) the coil geometry enables a higher applied field at the surface of the pipe compared to the standard coil configuration for the same power input, which should lead to a superior signal to noise ratio.

  • - 6) Minor comments: We have fixed the minor errors you identified and removed all the double spaces.

Reviewer 2 Report

Comments and Suggestions for Authors

In this manuscript, the authors present a method for detecting corrosion under insulation in pipes based on eddy current measurements. In specific, they aim to enhance the accuracy of the (well-established) system and measurement methodology by fine-tuning the parameters of the magnetic field sensors used in the detection process, overcoming SNR issues due to large distance from the pipe and alignment issues of the sensor relative to the radial magnetic field. In this context, the authors first perform simulations to investigate the effects of sensor angular alignment on the measured magnetic field, as well as the estimated defect depth. They then also illustrate the presented methodology using experimental measurements, targeting corrosion depth identification of 0.1 mm. The manuscript is well-written, interesting and relevant to the topics of Sensors. However, major revisions are required to enhance its quality:

There is something wrong with the Figures (probably a missing figure?) and most of the cross-references are wrong (also cross-references to Tables).

In line 384, you probably mean “The simulations from Section 2.1”?

Please state in the introduction what are the innovative aspects of your manuscript compared to previous work, since it is not clarified in the current version of the manuscript.

A photo from the experimental setup would significantly enhance the readability of section 3.

You mention in section 3.1.1. that: “A calibration process has been devised that allows both the cross-sensitivity and the sensor angle to be determined”. In my understanding, there are additional and complementary calibration techniques that can be used to enhance your presented methodology. There is no need to change your model in the current manuscript, but you should at least consider them and present them to the readers (maybe in the introduction section). Relevant references:

1.       Springmann, John, James Cutler, and Hasan Bahcivan. "Magnetic sensor calibration and residual dipole characterization for application to nanosatellites." AIAA/AAS Astrodynamics Specialist Conference. 2010.

2.       Kakarakis, Sarantis-Dimitrios J., et al. "A software-based calibration technique for characterizing the magnetic signature of EUTs in measuring facilities." IEEE Transactions on Electromagnetic Compatibility 59.2 (2016): 334-341.

3.       Bonnet, S., et al. "Calibration methods for inertial and magnetic sensors." Sensors and Actuators A: Physical 156.2 (2009): 302-311.

4.       Luo, Pinggui, et al. "Design and development of a self-calibration-based inductive absolute angular position sensor." IEEE Sensors Journal 19.14 (2019): 5446-5453.

Comments on the Quality of English Language

There are several typos throughout the manuscript, as well as syntax and grammatical errors (missing commas, strange syntax in some sentences, etc.) that make the paper hard to read, e.g., lines 30, 54, 102, 141, 147, 211, 270, 308-309, 335, 398, 412, 446, Table 3, 483, 494-495, 517-518.

Author Response

We thank the reviewers for their careful consideration of our manuscript and the useful suggestions made for improvements. Detailed responses to comments follow.

Reviewer 2

  • There is something wrong with the Figures (probably a missing figure?) and most of the cross-references are wrong (also cross-references to Tables).

Yes, sorry, there appears to have been an issue with the cross-referencing in the Word document. We have replaced these links with text, so this should no longer be an issue.  

  • In line 384, you probably mean “The simulations from Section 2.1”?

Yes, we have corrected the section reference.

  • Please state in the introduction what are the innovative aspects of your manuscript compared to previous work, since it is not clarified in the current version of the manuscript.

We have added a more explicit statement of the innovation aspects of this manuscript starting at line 77. The innovative aspects of this manuscript focus on the effect of sensor angle on defect detection for this type of eddy current system. An aspect not previously considered but shown in this paper has a considerable effect on corrosion detection.

  • A photo from the experimental setup would significantly enhance the readability of section 3.

We have added Figure 11, showing the experimental setup, as suggested.

  • You mention in section 3.1.1. that: “A calibration process has been devised that allows both the cross-sensitivity and the sensor angle to be determined”. In my understanding, there are additional and complementary calibration techniques that can be used to enhance your presented methodology. There is no need to change your model in the current manuscript, but you should at least consider them and present them to the readers (maybe in the introduction section).

We have reviewed your suggested references for the complementary calibration process and have referenced them as potential complementary calibration processes that could be used in the paragraph starting on line 318.

  • There are several typos throughout the manuscript, as well as syntax and grammatical errors (missing commas, strange syntax in some sentences, etc.) that make the paper hard to read, e.g., lines 30, 54, 102, 141, 147, 211, 270, 308-309, 335, 398, 412, 446, Table 3, 483, 494-495, 517-518.

We have reviewed the paper for syntax and grammar and fixed the errors found, thank you for the list provided.

Round 2

Reviewer 2 Report

Comments and Suggestions for Authors

In my view, the authors have responded to the raised comments, incorporating the appropriate modifications and significantly enhancing the quality of their manuscript, which I now recommend for publication in Sensors.